# Genomic Association of Single Nucleotide Polymorphisms with Blood Pressure Response to Hydrochlorothiazide among South African Adults with Hypertension

**DOI:** 10.3390/jpm10040267

**Published:** 2020-12-09

**Authors:** Charity Masilela, Brendon Pearce, Joven Jebio Ongole, Oladele Vincent Adeniyi, Mongi Benjeddou

**Affiliations:** 1Department of Biotechnology, University of the Western Cape, Bellville 7530, South Africa; brendon.biff@gmail.com (B.P.); mbenjeddou@uwc.ac.za (M.B.); 2Center for Teaching and Learning, Department of Family Medicine, Piet Retief Hospital, Mkhondo 2380, South Africa; jjongole@gmail.com; 3Department of Family Medicine, Walter Sisulu University, East London 5200, South Africa; vincoladele@gmail.com

**Keywords:** uncontrolled hypertension, hydrochlorothiazide, single nucleotide polymorphisms, pharmacogenomics, South Africa

## Abstract

This study described single nucleotide polymorphisms (SNPs) in hydrochlorothiazide-associated genes and further assessed their correlation with blood pressure control among South African adults living with hypertension. A total of 291 participants belonging to the Nguni tribes of South Africa on treatment for hypertension were recruited. Nineteen SNPs in hydrochlorothiazide pharmacogenes were selected and genotyped using MassArray. Uncontrolled hypertension was defined as blood pressure ≥140/90 mmHg. The association between genotypes, alleles and blood pressure response to treatment was determined by conducting multivariate logistic regression model analysis. The majority of the study participants were female (73.19%), Xhosa (54.98%) and had blood pressure ≥140/90 mmHg (68.73%). Seventeen SNPs were observed among the Xhosa tribe, and two (rs2070744 and rs7297610) were detected among Swati and Zulu participants. Furthermore, alleles T of rs2107614 (AOR = 6.69; 95%CI 1.42–31.55; *p* = 0.016) and C of rs2776546 (AOR = 3.78; 95%CI 1.04–13.74; *p* = 0.043) were independently associated with uncontrolled hypertension, whilst rs2070744 TC (AOR = 38.76; 95%CI 5.54–270.76; *p* = 0.00023), CC (AOR = 10.44; 95%CI 2.16–50.29; *p* = 0.003) and allele T of rs7297610 (AOR = 1.86; 95%CI 1.09–3.14; *p* = 0.023) were significantly associated with uncontrolled hypertension among Zulu and Swati participants. We confirmed the presence of SNPs associated with hydrochlorothiazide, some of which were significantly associated with uncontrolled hypertension in the study sample. Findings open doors for further studies on personalized therapy for hypertension in the country.

## 1. Introduction

Hypertension is the leading cause of death globally, accounting for 10.4 million deaths per year [1]. Furthermore, an estimated 1.13 billion people worldwide have been diagnosed with hypertension, and most reside in low-middle income countries. In South Africa, the highest rate of hypertension has been reported among individuals aged 50 years and above, with almost eight out of ten people in this age group having been diagnosed with high blood pressure [2]. In addition, the Heart and Stroke Foundation of South Africa reported that one in three South Africans 15 years and older are hypertensive [2]. The high burden of hypertension among South Africans is accompanied by low control rates as well as adverse cardiovascular disease risk [2,3]. While epidemiological studies have improved our understanding of the environmental factors associated with hypertension control, more especially with regards to physical activity and diet, the role of genetics in this setting remains unclear. Therefore, it is critical to explore genetic factors with regards to hypertension control in order to establish genetic-based initiatives that could be applied in medical practice to reduce the burden of hypertension and improve treatment outcomes among patients.

Hydrochlorothiazide (HCTZ) is a thiazide diuretic that is indicated for the treatment of hypertension [4]. The drug has been shown to lower blood pressure by acting on the kidneys to reduce sodium (Na^+^) reabsorption in the distal convoluted tubule [4,5]. Although HCTZ has been used as a first-line drug for the treatment of hypertension for over six decades, blood pressure response to the drug is highly variable [6,7]. As such, pharmacogenomics studies have investigated genetic polymorphisms that could account for the inter-individual variability that is observed across individual patients as well as diverse population groups. Single nucleotide polymorphisms (SNPs) in genes such as protein kinase C alpha (*PRKCA*), lysine deficient protein kinase 1 (*WNK1*) beta-2 adrenergic receptor (*ADRB2*) and nitric oxide synthase 3 (*NOS3*) have been of particular interest due to the role they play in blood pressure control [8,9,10].

The *PRKCA* gene encodes an enzyme that plays an important role in the modulation of ion channels [10]. In vivo studies suggest that this enzyme may be a fundamental regulator of cardiac contractility and Calcium (Ca^2+^) handling in myocytes [11]. On the other hand, the *WNK1* gene encodes for a ubiquitously expressed protein that regulates vasoconstriction and blood pressure response [12,13]. A study conducted among Caucasian hypertensive participants showed that an intronic SNP, rs16960228, in *PRKCA* is an important predictor of HCTZ blood pressure response. The study further demonstrated that rs16960228 A allele carriers had a greater blood pressure response compared to GG carriers [14]. In addition, hypertensive patients with the CC genotype of rs4791040 showed a greater reduction of diastolic blood pressure as compared to carriers of CT and TT genotypes following HCTZ treatment [14]. Inversely, hypertensive carriers of the CC genotype of rs2277869 (*WNK1*) showed increased ambulatory blood pressure as compared to carriers of CT and TT genotypes [8], whereas genotypes (CC and CT) of rs2107614 of the *WNK1* gene were associated with a greater reduction in whole-day ambulatory blood pressure among patients with essential hypertension who were treated with HCTZ [8].

The *ADRB2* and *NOS3* genes are central components of the renin–angiotensin system (RAS) that controls blood pressure by regulating the volume of fluids in the body [7,15]. As such, polymorphisms in these genes might influence blood pressure control. A study conducted in a cohort comprised of 50% individuals of African origin showed that the AA and AG genotypes of rs2400707 (*ADRB2*) were associated with an increased reduction in whole-day ambulatory blood pressure following hydrochlorothiazide treatment [8]. On the other hand, it was shown that hypertensive carriers of the CC genotype of rs2070744 (*NOS3*) who were treated with anti-hypertensive drugs including diuretics may have an increased risk for resistance to medication as compared to patients with the CT or TT genotype [16]. However, a direct association of the genotypes of rs2070744 (*NOS3*) with blood pressure response to HCTZ is yet to be established.

The *YEATS4* gene encodes the protein GAS41, which has been shown to mediate RNA transcription and cell viability [17]. Unlike the *ADRB2, PRKCA* and *WNK1* genes, there was no reference found in the literature that connects the *YEATS4* gene with pathways associated with hypertension or drug response. However, carriers of rs7297610 (CC genotype) were associated with greater blood pressure responses to HCTZ in comparison to T allele carriers. It was further demonstrated that such an association was absent among atenolol-treated participants [18]. Therefore, these findings suggest that there could be a potential mechanism where *YEATS4* could affect blood pressure response to thiazide diuretic medication. However, further research is needed to verify this association.

Pharmacogenomics has progressed and matured into an efficient and effective tool for mapping genes underlying human phenotypes associated with drug response. This tool holds the promise of using genome-based technologies to improve health by effectively treating diseases including hypertension [19,20,21,22]. However, pharmacogenomics is still in its infancy in the developing world, and little is known about the influence of genetic factors on blood pressure response to hypertensive treatment among Africans. In this study, we described single nucleotide polymorphisms (SNPs) in hydrochlorothiazide-associated genes and further assessed their correlation with blood pressure control among South African adults living with hypertension.

## 2. Materials and Methods

### 2.1. Ethical Approval

The Senate Research Committee of the University of the Western Cape approved the study protocol (Ethics approval number: BM/16/5/19). Permission to implement the study was granted by the clinical governance of the respective hospitals in the Eastern Cape and Mpumalanga Provinces. Participants were issued with a research information sheet detailing the study, and it was made available in three indigenous languages (SiSwati, IsiXhosa and IsiZulu). Each participant indicated their voluntary participation by signing a consent form. The rights to privacy and confidentiality of the medical information of each participant were honored during and after the study.

### 2.2. Patient Selection

A total of 291 Nguni (Xhosa, Swati and Zulu) patients attending chronic care for hypertension were recruited consecutively between January 2019 and June 2019 from Cecilia Makiwane Hospital (East London, Eastern Cape), Piet Retief Hospital, Thandukukhanya Community Health Center and Mkhondo Town Clinic (Mkhondo, Mpumalanga). Participants were eligible for participation if they were 18 years or older and were on continuous treatment for hypertension for at least a year prior to the study. Individuals who were bedridden, pregnant and unable to give consent were excluded from the study.

### 2.3. Data Collection

A trained research nurse measured the blood pressure (BP) of each participant by using a validated automated digital BP monitor (Macrolife BP A 100 Plus model) according to standard protocols. Thereafter, BP was recorded in triplicate, and the average was used to categorize participants into two groups: controlled (blood pressure < 140/90 mmHg) and uncontrolled (blood pressure ≥ 140/90 mmHg). DNA samples were collected in the form of buccal swabs and stored at −20 °C until they were processed.

The age, ethnicity, smoking status and salt intake were self-reported by each participant and documented in a proforma designed for this study. The number and type of anti-hypertensive drugs prescribed for each participant were retrieved from their clinical records.

### 2.4. DNA Isolation

Genomic DNA was extracted from buccal swab samples using a standard salt-lysis procedure. Briefly, DNA samples were incubated in lysis buffer at 62 °C overnight. Thereafter, DNA was precipitated with NaCl followed by the addition of 75% ice-cold ethanol and incubated at −20 °C overnight. Precipitated DNA was purified using 70% ethanol and re-suspended in nuclease-free water. Samples were stored in 2 mL Eppendorf tubes at −20 °C until further use. DNA was quantified using a NanoDrop™ 2000/2000c spectrophotometer (Thermo Scientific™, Waltham, MA, USA) and Gel Doc™ EZ Gel Documentation System (BIO-RAD, Irvine, CA, USA).

### 2.5. Genotyping

Two multiplex MassARRAY systems (Agena Bioscience^TM^) were designed and optimized by Inqaba Biotechnical Industries (Pretoria, South Africa) in January 2017. Each multiplex was used to genotype selected SNPs, using an assay that is based on a locus-specific PCR reaction. This reaction is followed by a single base extension using the mass-modified dideoxynucleotide terminators of an oligonucleotide primer, which anneals upstream of the site of mutation. Matrix-Assisted Laser Desorption/Ionization–time-of-flight (MALDI-TOF) mass spectrometry was used to identify the SNP of interest.

### 2.6. Statistical Analysis

Statistical analyses were performed using International Business Machines (IBM) Statistical Package for Social Science (SPSS) Version 25 for Windows (IBM Corps, Armonk, NY, USA). The general characteristics of the participants were expressed as frequency (percentages). The associations between alleles, genotypes and blood pressure response to hydrochlorothiazide were assessed by multivariate logistic regression model analysis (unadjusted and adjusted odds ratios) and their 95% confidence intervals (95%CI). The final model of the adjusted logistic regression analysis for the Xhosa population included rs11189015, rs1458038, rs16960228, rs17010902, rs2106809, rs2107614, rs2269879, rs2277869, rs2400707, rs2776546, rs292449, rs4149601, rs4551053, rs4791040 and rs5051. For the Swati and Zulu population, the final adjusted regression model analysis included rs6083538, rs2070744 and rs7297610. Results for the unadjusted logistic regression model analysis were expressed as unadjusted odds ratios (ORs) and adjusted odds ratios (AORs) for the adjusted logistic regression model analysis. A *p*-value less than 0.05 was considered statistically significant. Bonferroni corrected *p*-values were set at <0.0029 for the Xhosa population and <0.025 for the Swati and Zulu population. Minor allele frequency (MAF) and Hardy–Weinberg equilibrium (HWE) tests were calculated using Genetic Analysis in Excel (GenAIEx) Version 6.5.

### 2.7. Selection of Pharmacogenomics Biomarkers

Nineteen SNPs previously associated with hypertension or hydrochlorothiazide efficacy were selected using Pharmacogenomics Knowledge Base Ensembl [23] as well as an extensive survey of recent literature. Selected SNPs were in genes that are indirectly or directly involved in the pathways associated with the blood-pressure-lowering effect of hydrochlorothiazide on hypertension exhibiting a PharmGKB evidence rating of at least 3.

## 3. Results

### 3.1. General Characteristics of the Study

A total of 291 individuals with hypertension participated in this study, of whom 73.19% (*n* = 213) were female and 26.04% (*n* = 78) were male. The mean age (SD) of the participants was 60.45 ± 11.90 years. The cohort was composed of individuals belonging to the Xhosa (*n* = 160), Zulu (*n* = 112) and Swati (*n* = 19) tribes of South Africa. The majority of the participants were non-smokers (67.35%), consumed low-moderate salt (81.44%) and had blood pressure ≥140/90 mmHg (68.73%) (Table 1).

### 3.2. Expression Patterns of Single Nucleotide Polymorphisms

Nineteen SNPs were selected and their expression patterns were assessed across three populations (Swati, Xhosa and Zulu). Seventeen out of nineteen SNPs were exclusively detected among the Xhosa tribe (*n* = 160), the remaining two (rs2070744 and rs7297610) were detected among Swati and Zulu participants. The majority of the seventeen SNPs detected among the Xhosa tribe demonstrated an expression frequency above 90%, with variants rs4791040 and rs5051 showing an expression frequency of 73.10% (*n* = 117) and 68.75% (*n* = 110), respectively. Variant rs2070744 and rs5051 showed a 100% expression efficiency among Swati (*n* = 19) and Zulu (*n* = 112) participants, of whom 51.17% (*n* = 109) were female, and 44.44% (*n* = 40) were aged 60 years and above (Table 2). The minor allele frequency (MAF) observed in all three populations was compared to the Luhya people of Kenya, the Yoruba of Nigeria, Mexican from California (USA), British of Great Britain and Punjabi of India. Variants rs11189015 (33.5%), rs17010902 (59.5%), rs2106809 (88.5%) and rs2277869 (20.5%) detected among the Xhosa tribe showed a higher MAF in comparison to the selected reference populations listed on Ensembl (23). However, the MAFs of rs2269879 (32.2%), rs2400707 (37.26%) and rs1458038 (20.3%) were lower than those observed in the selected world populations. The MAFs of the remaining SNPs are shown in Table 3. Variant rs7297610 (52.4%) detected among the Swati and Zulu tribe showed a higher MAF when compared to the selected world populations. The MAF observed in variant rs2070744 (14.7%) was lower than the MAF observed among British, Mexican and Punjabi populations (Table 3). None of the SNPs in this cohort deviated from the Hardy–Weinberg equilibrium.

### 3.3. Association between SNPs and Blood Pressure Response to Hydrochlorothiazide

In the multivariate logistic regression (unadjusted) model analysis, the allele A of rs2400707 (OR = 7.34; 95%CI 3.05–17.67; *p* ≤ 0.0001) was independently associated with uncontrolled hypertension, although carriers of the genotype of rs2400707 AA (OR = 0.36; 95% CI 0.15–0.85; *p* = 0.020) were less likely to have uncontrolled hypertension. No association was established in the remaining sixteen SNPs detected among the Xhosa tribe. In the adjusted logistic regression model, the direction of association for the A allele of rs2400707 shifted, and carriers of the A allele (OR = 0.14; 95%CI 0.03–0.665; *p* = 0.013) were less likely to have uncontrolled hypertension. Furthermore, carriers of allele T of rs2107614 (AOR = 6.69; 95%CI 1.42–31.55; *p* = 0.016) and C of rs2776546 (AOR = 3.78; 95%CI 1.04–13.74; *p* = 0.043) were almost seven times and four times more likely to have uncontrolled hypertension, respectively. On the other hand, carriers of rs4791040 allele C (AOR = 0.10; 95%CI 0.01–0.60; *p* = 0.012) were less likely to have uncontrolled hypertension. After Bonferroni correction, all the alleles of rs2400707 (A), rs2107614 (T), rs2776546 (C) and rs4791040 remained significant with *p*-values < 0.0029 (Table 4).

Among Zulu and Swati participants, the multivariate logistic regression model analysis showed that carriers of the genotype CC of rs2070744 (OR = 4.22; 95%CI 1.15–15.47; *p* = 0.030) were four times more likely to be associated with uncontrolled hypertension, whilst rs2070744 TC (OR = 0.10; 95%CI 0.02–0.48; *p* = 0.004), rs7297610 CT (OR = 0.40; 95%CI 0.16–0.98; *p* = 0.045) and allele T (OR = 0.60; 95%CI 0.36–0.98; *p* = 0.043) carriers were less likely to have uncontrolled hypertension. After adjusting with each SNP, genotypes rs2070744 TC (AOR = 38.76 95%CI 5.54–270.76; *p* = 0.003) and CC (AOR = 10.44; 95%CI 2.16–50.29; *p* = 0.00023) were significantly associated with uncontrolled hypertension. In addition, allele T of rs7297610 (AOR = 1.86; 95%CI 1.09–3.14; *p* = 0.023) was independently associated with uncontrolled hypertension. After Bonferroni correction, the genotypes of rs2070744 and T allele of rs7297610 remained significantly associated with uncontrolled hypertension (*p* < 0.0025) (Table 5).

## 4. Discussion

Thiazide diuretics are among the most prescribed anti-hypertensive drugs worldwide [24]. Furthermore, this class of drugs is recommended for the initial treatment of hypertension [4]. However, pharmacogenetic markers of thiazide efficacy among African-specific populations are not well studied. As such, there is a huge knowledge gap on the effect of SNPs and blood pressure response to thiazide diuretics among populations of African origin. Therefore, this study described single nucleotide polymorphisms (SNPs) in hydrochlorothiazide-associated genes and further assessed their correlation with blood pressure control among South African adults living with hypertension.

Current research suggests that the genomes of indigenous African individuals carry the greatest depth of genetic variation compared to other population groups from around the world [25]. Thus, studying African-specific populations could help researchers understand drug response phenotypes in order to improve treatment outcomes for people living with hypertension. In the current study, nineteen SNPs previously associated with hydrochlorothiazide efficacy in individuals with hypertension were examined in 291 individuals belonging to the Zulu, Xhosa and Swati tribes (Nguni) of South Africa. Seventeen SNPs were detected among the Xhosa tribe, and only two SNPs (rs2070744 and rs7297610) were detected among the Swati and Zulu people. The minor allele frequencies of rs17010902, rs11189015, rs2277869 and rs2106809 were particularly higher among the Xhosa tribe when compared to other populations (Yoruba, Luhya, Mexican, British and Punjabi), whilst rs6083538 showed a lower minor allele frequency when compared to non-African populations (Mexican, British and Punjabi). The minor allele frequencies of the remainder of SNPs were comparable to the selected African populations (Yoruba and Luhya) as well as those from other parts of the world. In addition, the minor allele frequencies of the two SNPs detected among the Swati and Zulu people were also compared with other population groups. Variant rs7297610 showed a higher minor allele frequency in comparison to all the other population groups. Variant rs2070744 demonstrated a frequency similar to that of Luhya people (Kenya), however, lower than the minor allele frequencies observed across Mexican, British and Punjabi population groups. The genetic architecture of Nguni-speaking tribes has been described as fairly homogeneous, however, the finding of this study suggests that some disparities in blood pressure response to hydrochlorothiazide brought by SNPs that each tribe possesses may exist. Although this panel of SNPs does not represent the entire human genome, it at least opens doors for more genetic studies in order to gain a broader understanding of personalized treatment in patient care, especially in individuals with hypertension. Findings from future studies with a larger sample size drawn from the broader ethnically diverse population of South Africans might guide the selection and dosing of thiazide diuretics as well as other hypertensive drugs.

The *WNK1* gene encodes a protein that plays an important role in renal ion transport [13]. On the other hand, the *ADRB2* gene mediates a rise in intracellular cAMP concentration, which, through smooth muscle relaxation, leads to vasodilation [4]. Blunted ADRB2 and WNK1 function have been implicated in the pathogenesis of hypertension. In this study, the T allele of rs2107614 (*WNK1*) was significantly associated with uncontrolled blood pressure among Xhosa participants, however, no association was established with any of the genotypes. In contrast, Turner et al. (2005) showed that the genotypes CC and CT of rs2107614 (WNK1) were associated with an increased reduction in whole-day ambulatory blood pressure among individuals with non-complicated hypertension treated with HCTZ [8]. On the other hand, this study showed that carriers of the A allele and the AA genotype of rs2400707 (*ADRB2*) were less likely to have uncontrolled blood pressure. These observations are in line with previous findings, where the AA and AG genotypes of rs2400707 (*ADRB2*) were associated with an increased reduction in whole-day ambulatory blood pressure in individuals with essential hypertension undergoing HCTZ treatment [8]. These findings indicate that polymorphisms in genes regulating renal sodium transport and smooth muscle relaxation may predict inter-individual variability in blood pressure response to HCTZ. Furthermore, these genes as well as their SNPs may serve as therapeutic markers for individualizing thiazide treatment for hypertensive patients of African ancestry.

The variant rs2776545 occurs in the regulatory region of the CUB and Sushi multiple domains 1 (*CSMD1*) that encodes a product that functions as a complement control protein [26]. In this study, allele C of rs2776546 was associated with uncontrolled blood pressure among patients belonging to the Xhosa tribe. However, no association was established between HCTZ treatment response and the genotypes of the SNPs. Conversely, a previous study showed that the A allele of rs2776546 was associated with increased response to thiazide diuretics in people with hypertension as compared to allele C. It was further demonstrated that carriers of the AA genotype of European ancestry treated with HCTZ showed a greater reduction of diastolic blood pressure as compared to patients with the AC or CC genotypes [14]. Moreover, the *CSMD1* gene was associated with an increased risk of hypertension among Korean patients [27,28]. Although the role of *CSMD1* in the pathophysiology of hypertension is not completely understood, the findings of this study bring attention to clinically relevant loci of blood pressure response to thiazide diuretics among individuals of African ancestry and further highlight the need for more studies with larger sample sizes that could validate the direction of association of each allele and genotype.

The *PRKCA* gene is an important regulator of many physiological functions including secretion and exocytosis, modulation of ion channel (Ca^2+^ ions) gene expression and cell growth and proliferation [29] that harbors the SNP rs4791040. The current study showed that Xhosa carriers of the C allele of rs4791040 were less likely to have uncontrolled blood pressure. Furthermore, a previous study conducted among hypertensive patients of European origin showed that the allele T of rs4791040 was associated with decreased response to diuretics including hydrochlorothiazide as compared to allele C. It was further demonstrated that carriers of TT genotype treated with HCTZ may have a decreased reduction of diastolic blood pressure as compared to patients with the CC or CT genotypes [8]. Although no association was established between the genotypes of rs4791040 and blood pressure response to hydrochlorothiazide in the present study, the current findings provide substantial evidence that *PRKCA* polymorphisms may influence blood pressure response to hydrochlorothiazide owing to their role in the modulation of ion channels.

Single nucleotide polymorphism rs2070744 is an intronic variant that sits on the *NOS3* gene. In addition, rs2070744 has been implicated in the variable response of thiazide diuretics. Carriers of the CC genotype treated with anti-hypertensive drugs including HCTZ demonstrated an increased risk of resistant hypertension as compared to TC and TT carriers [30]. It was further demonstrated that carriers of the TC genotype may have a decreased, but not absent, risk for resistant hypertension. Moreover, the authors described resistant hypertension as uncontrolled blood pressure when treated with lifestyle measures and at least three anti-hypertensive drugs at maximum doses including a diuretic. In the present study, uncontrolled hypertension was defined as blood pressure ≥140/90 mmHg whilst on treatment. It should, however, be noted that lifestyle behaviors, doses of anti-hypertensive drugs and the effect of individual drugs were not quantified in this study. The degree of association between HCTZ treatment response and SNPs was solely measured without taking into consideration other drugs administered. Furthermore, carriers of CC and TC genotypes (rs2070744) were more likely to have uncontrolled hypertension. However, no clear association was established between the alleles of rs2070744 and blood pressure response to HCTZ. Although the findings suggest a significant association between the CC genotype and blood pressure response to hydrochlorothiazide, the large difference between the numbers of alleles observed discounts this significance. Given the large difference in numbers between the T allele and C allele, coupled with the uneven spread of genotypes at this locus, it may be suggested that these findings are not significant but are instead a product of skewed observations. Larger samples are required to definitively establish the observed association in this study.

This study also investigated the effect of *YEATS4* polymorphism (rs7297610) on blood pressure response to hydrochlorothiazide. The T allele of rs7297610 was independently associated with uncontrolled hypertension among Swati and Zulu patients. This study further demonstrated that carriers of CT genotype were less likely to have uncontrolled blood pressure. The observations made in this study are in line with previous findings, where allele C was associated with an increased reduction in blood pressure among individuals of mixed ancestry (African American and Afro-Caribbean) treated with hydrochlorothiazide as compared to allele T [18]. Although other genetic and clinical factors may also influence a patient’s response to hydrochlorothiazide, the study further demonstrated that patients with the TT genotype treated with hydrochlorothiazide may have a decreased response as compared to patients with the CC genotype [18]. Additionally, a haplotype made from three SNPs, rs317689/rs315135/rs7297610 (ATC), was strongly associated with greater HCTZ response with the ACT and ATT haplotypes correlating with a smaller blood pressure response [31]. Of note, the role of YEATS4 in the development of hypertension remains elusive, however, previous findings and observations made in this study suggest that polymorphism in this gene may predict blood pressure response to thiazide diuretics among patients of African ancestry. It is also possible that the effect of rs7297610 on HCTZ blood pressure response is a result of an interaction with other functional SNPs not yet known. As such, more studies need to be conducted in order to explore the functional role of YEATS4 and the mechanism in which it affects blood pressure in response to thiazide diuretics.

## 5. Conclusions

Using a candidate gene approach, we identified seventeen SNPs among the Xhosa tribe and two SNPs among the Zulu and Swati tribes previously associated with hydrochlorothiazide efficacy and hypertension. The minor alleles of rs2107614 and rs2776546 were independently associated with uncontrolled hypertension among Xhosa participants. Furthermore, the T allele of rs7297610 was independently and significantly associated with uncontrolled hypertension among Swati and Zulu participants. This study also provided preliminary information for the association of *YEATS4* polymorphisms in blood pressure response to hydrochlorothiazide. However, replication of these findings in a larger South African cohort is needed to confirm the associations observed in this study. Further elucidation of the exact mechanism in which these SNPs affect blood pressure in response to hydrochlorothiazide can ultimately aid in improving individualized anti-hypertensive therapy and the identification of new drug targets.

## Figures and Tables

**Table 1 jpm-10-00267-t001:** General characteristics of the study cohort. HCTZ: hydrochlorothiazide.

Variables	All Participants (*n*; %)	Males (*n*; %)	Females (*n*; %)
All	291(100)	78(26.04)	213(73.19)
Age (Years)			
18–25	01(0.34)	01(1.28)	0(0.00)
26–35	08(2.75)	02(2.56)	06(2.82)
36–45	19(6.52)	07(8.97)	12(6.63)
46–55	52(17.87)	14(17.95)	38(17.84)
56–65	120(41.24)	25(32.05)	95(44.60)
≥66	91(31.27)	29(37.18)	62(29.11)
Ethnicity			
Zulu	112(38.49)	19(24.36)	93(43.66)
Swati	19(6.53)	03(3.85)	16(7.51)
Xhosa	160(54.98)	56(71.79)	104(48.83)
Smoking status			
Never smoked	196(67.35)	30(38.46)	166(77.93)
Ever smoked	95(32.65)	48(61.54)	47(22.07)
Salt intake			
Low-moderate	237(81.44)	58(74.36)	179(84.04)
Increased	54(18.56)	20(25.64)	34(15.96)
Blood pressure			
<140/90 mmHg	91(31.26)	16(20.51)	75(35.21)
≥140/90 mmHg	200(68.73)	62(79.49)	138(64.79)
Drug regime			
HCTZ alone	63(21.65)	20(25.64)	43(20.19)
HCTZ + 1 drug	127(43.64)	26(33.33)	101(47.42)
HCTZ+ 2 drugs	98(33.68)	30(38.46)	68(31.92)
HCTZ + 3 drugs	03(1.03)	02(2.56)	01(0.47)

Anti-hypertensive drugs used in different combinations: Amlodipine, Enalapril and Atenolol.

**Table 2 jpm-10-00267-t002:** Distribution patterns of selected single nucleotide polymorphisms (SNPs).

dbSNP	Gene	Ethnic Groups			Gender		Age		
		Zulu (*n*; %)	Swati (*n*; %)	Xhosa (*n*; %)	Male (*n*; %)	Female (*n*; %)	<55 Years	55–65 Years	>65 Years
All		112(38.48)	19(6.52)	160(54.98)	78(26.80)	213(73.19)	80(27.49)	121(41.58)	90(30.93)
rs11189015	*SLIT1*								
Yes		-	-	155(96.88)	54(69.23)	101(47.42)	43(53.75)	62(51.24)	50(55.56)
No		112(100)	19(100)	05(3.10)	24(30.77)	112(52.58)	37(46.25)	59(48.76)	40(44.44)
rs1458038	*FGF5*								
Yes		-	-	156(97.50)	55(70.51)	101(47.42)	45(56.25)	62(51.24)	49(54.44)
No		112(100)	19(100)	04(2.50)	23(29.49)	112(52.58)	35(43.75)	59(48.76)	41(45.56)
rs16960228	*PRKCA*								
Yes		-	-	158(98.75)	56(71.79)	102(47.89)	47(58.75)	63(52.07)	48(53.33)
No		112(100)	19(100)	02(1.25)	22(28.21)	111(52.11)	33(41.25)	58(47.93)	42(46.67)
rs17010902	*APOA5*								
Yes		-	-	152(95.00)	54(69.23)	98(46.01)	44(55.00)	61(50.41)	47(52.22)
No		112(100)	19(100)	08(5.00)	24(30.77)	115(53.99)	36(45.00)	60(49.59)	43(47.78)
rs2106809	*ACE2*								
Yes		-	-	157(98.10)	54(69.23)	103(48.36)	45(56.25)	62(51.24)	50(55.56)
No		112(100)	19(100)	03(1.90)	24(30.77)	110(51.64)	35(43.75)	59(48.76)	40(44.44)
rs2107614	*WNK1*								
Yes		-	-	155(96.90)	55(70.51)	100(46.95)	47(58.75)	62(51.24)	46(51.11)
No		112(100)	19(100)	05(3.10)	23(29.49)	113(53.05)	33(41.25)	59(48.76)	44(48.89)
rs2269879	*DOT1L*								
Yes		-	-	156(97.50)	54(69.23)	102(47.89)	44(55.00)	63(52.07)	49(54.44)
No		112(100)	19(100)	04(2.50)	24(30.77)	111(52.11)	36(45.00)	58(47.93)	41(45.56)
rs2277869	*WNK1*								
Yes		-	-	156(97.50)	55(70.51)	101(47.42)	46(57.50)	61(50.41)	49(54.44)
No		112(100)	19(100)	04(2.50)	23(29.49)	112(52.58)	34(42.50)	60(49.59)	41(45.56)
rs2400707	*ADRB2*								
Yes		-	-	157(98.10)	56(71.79)	101(47.42)	45(56.25)	63(52.07)	49(54.44)
No		112(100)	19(100)	03(1.90)	22(28.21)	112(52.58)	35(43.75)	58(47.93)	41(45.56)
rs2776546	*CSMD1*								
Yes		-	-	158(98.75)	56(71.79)	102(47.89)	47(58.75)	63(52.07)	48(53.33)
No		112(100)	19(100)	02(1.25)	22(28.21)	111(52.11)	33(41.25)	58(47.93)	42(46.67)
rs292449	*NEDD4L*								
Yes		-	-	156(97.50)	55(70.51)	101(47.42)	45(56.25)	63(2.07)	48(53.33)
No		112(100)	19(100)	04(2.50)	23(29.49)	112(52.58)	35(43.75)	58(47.93)	42(46.67)
rs3184504	*SH2B3*								
Yes		-	-	159(99.40)	56(71.79)	103(48.36)	47(58.75)	63(52.07)	49(54.44)
No		112(100)	19(100)	01(0.60)	22(28.21)	110(51.64)	33(41.25)	58(47.93)	41(45.56)
rs4149601	*NEDD4L*								
Yes		-	-	159(99.40)	56(71.79)	103(48.36)	47(58.75)	63(52.07)	49(54.44)
No		112(100)	19(100)	01(0.60)	22(28.21)	110(51.64)	33(41.25)	58(47.93)	41(45.56)
rs4551053	*EBF1*								
Yes		-	-	160(100)	56(71.79)	104(48.83)	47(58.75)	63(52.07)	50(55.56)
No		112(100)	19(100)	-	22(28.21)	109(51.17)	33(41.25)	58(47.93)	40(44.44)
rs4791040	*PRKCA*								
Yes		-	-	110(68.75)	38(48.72)	72(33.80)	30(37.50)	42(34.71)	38(42.22)
No		112(100)	19(100)	50(31.25)	40(51.28)	141(66.20)	50(62.50)	79(65.29)	52(57.78)
rs5051	*AGT*								
Yes		-	-	117(73.10)	41(52.56)	76(35.68)	31(38.75)	48(39.67)	38(42.22)
No		112(100)	19(100)	43(26.90)	37(47.44)	137(64.32)	49(61.25)	73(60.33)	52(57.78)
rs6083538	*ZNF343*								
Yes		-	-	156(97.50)	56(71.79)	100(46.95)	46(57.50)	63(52.07)	47(52.22)
No		112(100)	19(100)	04(2.50)	22(28.21)	113(53.05)	34(42.50)	58(47.93)	43(47.78)
rs2070744	*NOS3*								
Yes		112(100)	19(100)	-	22(28.21)	109(51.17)	33(41.25)	58(47.93)	40(44.44)
No		-	-	160(100)	56(71.79)	104(48.83)	47(58.75)	63(52.07)	50(55.56)
rs7297610	*YEATS4*								
Yes		112(100)	19(100)	-	22(28.21)	109(51.17)	33(41.25)	58(47.93)	40(44.44)
No		-	-	160(100)	56(71.79)	104(48.83)	47(58.75)	63(52.07)	50(55.56)

**Table 3 jpm-10-00267-t003:** Minor allele frequency distribution across different population groups. MAF: minor allele frequency.

dbSNP	Nucleotide Substitution	Feature				MAF (%)			
			Xhosa	Swati and Zulu	Yoruba	Luhya	Mexican	British	Punjabi
rs11189015	C > G	Intron	33.5	-	29.6	29.8	3.9	6.6	14.1
rs1458038	C > T	Intergenic	20.3	-	94.4	97.5	73.4	73.6	77.1
rs16960228	C > T	Intron	2.2	-	40.7	28.8	9.4	7.7	0.5
rs17010902	A > G	Intergenic	59.5	-	0.5	3.5	26.0	8.8	16.1
rs2106809	A > G	Intron	88.5	-	7.3	7.8	35.0	25.7	40.3
rs2107614	T > C	Intron	53.5	-	38.9	53.0	64.8	73.1	69.3
rs2269879	C > T	Intron	32.2	-	64.4	63.1	75.8	93.4	91.1
rs2277869	T > C	Non-coding exon	20.5	-	13.0	17.7	18.0	14.3	19.3
rs2400707	A > G	5 prime UTR	37.26	-	56.9	43.3	82.8	60.4	70.8
rs2776546	C > A	Regulatory region	48.7	-	41.2	40.4	26.6	17.0	13.0
rs292449	G > C	5 prime UTR	49.0	-	45.4	47.0	46.9	70.3	56.8
rs3184504	C > G	Missense	100	-	100	100	22.7	50.0	7.3
rs4149601	G > A	Splice region	51.5	-	37.5	50.0	11.7	31.3	18.8
rs4551053	G > A	Regulatory region	13.6	-	11.1	11.6	16.4	33.0	43.2
rs4791040	T > C	Intron	38.6	-	40.7	28.8	10.7	7.7	3.1
rs5051	C > T	Intron	95.7	-	94.4	89.4	68.8	36.8	62.0
rs6083538	C > T	Intron	15.0	-	7.9	9.1	46.9	47.8	37.5
rs2070744	C > T	Intron	-	14.7	12.5	14.1	27.3	46.3	25.5
rs7297610		Intergenic	-	52.4	37.0	31.3	5.5	6.0	2.1

**Table 4 jpm-10-00267-t004:** Adjusted and unadjusted logistic regression models showing genotypes and alleles associated with blood pressure response to hydrochlorothiazide among Xhosa patients (*n* = 160).

dbSNP	Uncontrolled HPT (*n*; %)	Controlled HPT (*n*; %)	Unadjusted Odds Ratios (95% CI)	*p*-Value	Adjusted Odds Ratios (95% CI)	*p*-Value	Bonferroni-Adjusted *p*-Values
All	31(19.37)	129(80.63)					
rs11189015							
Genotypes							
CC	11(84.62)	02(15.38)	1		1		
GG	52(76.47)	16(23.53)	1.01(0.25–4.03)	0.982	0.82(0.30–2.22)	0.699	0.041
CG	62(83.78)	12(16.22)	1.14(0.49–2.62)		0.78(0.14–4.35)	0.786	0.046
Alleles							
G	166(79.05)	44(20.95)	1		1		
C	84(84.00)	16(16.00)	1.01(0.55–1.83)	0.969	1.65(0.40–6.70)	0.484	0.028
rs1458038							
Genotypes							
CC	89(81.65)	20(18.35)	1		1		
TT	06(66.67)	03(33.33)	0.85(0.34–2.10)	0.727	2.32(0.32–16.87)	0.403	0.023
CT	39(86.67)	06(13.33)	0.47(0.10–2.25)	0.352	1.59(0.24–10.18)	0.622	0.036
Alleles							
C	217(82.51)	46(17.49)	1		1		
T	51(80.95)	12(19.05)	1.02(0.50–2.08)	0.939	0.85(0.19–3.74)	0.832	0.048
rs16960228							
Genotypes							
CC	123(81.46)	28(18.54)	1		1		
TT	-	-	-		-		
TC	04(57.14)	03(42.86)	2.19(0.51–9.32)	0.286	0.29(0.46–1.91)	0.201	0.011
Alleles							
T	04(57.14)	03(42.86)	1		1		
C	250(80.91)	59(19.09)	1.42(0.54–3.76)	0.470	4.11(0.74–22.58)	0.104	0.006
rs17010902							
Genotypes							
GG	44(86.27)	07(13.73)	1		1		
AA	17(77.27)	05(22.73)	1.59(0.63–3.99)		0.45(0.98–2.12)	0.318	0.018
AG	61(77.22)	18(22.78)	1.33(0.44–4.01)		0.46(0.06–3.67)	0.471	0.027
Alleles							
A	95(77.24)	28(22.76)	1		1		
G	149(82.32)	32(17.68)	1.37(0.77–2.42)	0.275	0.34(0.57–2.10)	0.249	0.014
rs2106809							
Genotypes							
GG	09(75.00)	03(25.00)	1		1		
AA	109(81.95)	24(18.05)	0.75(0.13–4.25)	0.745	0.52(0.88–3.08)	0.472	0.027
AG	09(75.00)	03(25.00)	1.36(0.34–5.32)	0.657	0.40(0.75–2.16)	0.290	0.017
Alleles							
G	27(75.00)	09(25.00)	1		1		
A	227(81.65)	51(18.35)	1.48(0.65–3.34)	0.342	0.73(0.10–5.04)	0.752	0.044
rs2107614							
Genotypes							
CC	24(80.00)	06(20.00)	1		1		
TT	33(80.49)	08(19.51)	1.06(0.38–3.00)	0.901	1.17(0.37–3.68)	0.786	0.046
TC	58(78.38)	16(21.62)	1.04(0.41–2.66)	0.923	1.04(0.23–4.57)	0.952	0.056
Alleles							
C	106(79.10)	28(20.90)	1		1		
T	124(79.49)	32(20.51)	1.01(0.57–1.77)	0.970	6.69(1.42–31.55)	0.016	0.0009
rs2269879							
Genotypes							
CC	60(83.33)	12(16.67)	1		1		
TT	12(75.00)	04(25.00)	1.48(0.64–3.45)	0.357	0.63(0.11–3.44)	0.599	0.035
CT	53(77.94)	15(22.06)	0.92(0.26–3.23)	0.896	1.09(0.19–6.16)	0.918	0.054
Alleles							
C	173(81.60)	39(18.40)	1		1		
T	77(77.00)	23(23.00)	0.75(0.42–1.34)	0.342	1.49(0.33–6.65)	0.597	0.035
rs2277869							
Genotypes							
CC	03(75.00)	01(25.00)	1		1		
TT	77(80.21)	19(19.79)	0.70(0.06–7.41)	0.769	0.46(0.02–7.88)	0.599	0.035
CT	45(80.36)	11(19.64)	0.97(0.42–2.22)	0.948	1.06(0.38–2.92)	0.899	0.052
Alleles							
C	51(79.69)	13(20.31)	1		1		
T	199(80.24)	49(19.76)	1.03(0.52–2.05)	0.921	0.32(0.06–1.64)	0.174	0.010
rs2400707							
GG	37(68.52)	17(31.48)	1		1		
AA	22(88.00)	03(12.00)	0.36(0.15–0.85)	0.020	0.84(0.16–4.28)	0.842	0.049
AG	67(85.90)	11(14.10)	1.24(0.31–4.83)	0.757	0.27(0.05–1.30)	0.105	0.006
Alleles							
G	141(75.81)	45(24.19)	1		1		
A	111(86.72)	17(13.28)	7.34(3.05–17.67)	<0.0001	0.14(0.03–0.66)	0.013	0.0007
rs2776546							
Genotypes							
AA	32(82.05)	07(17.95)	1		1		
CC	29(82.86)	06(17.14)	1.28(0.48–3.38)	0.611	0.91(0.28–2.90)	0.878	0.051
CA	66(78.57)	18(21.43)	1.36(0.49–3.78)	0.551	1.12(0.28–4.40)	0.862	
Alleles							
A	130(80.25)	32(19.75)	1		1		
C	124(80.52)	30(19.48)	1.01(0.58–1.77)	0.951	3.78(1.04–13.74)	0.043	0.0025
rs292449							
Genotypes							
GG	29(74.36)	10(25.64)	1		1		
CC	35(83.33)	07(16.67)	0.68(0.27–1.73)	0.427	0.80(0.24–2.68)	0.723	0.043
GC	61(81.33)	14(18.67)	1.24(0.46–3.36)	0.665	0.59(0.16–2.41)	0.422	
Alleles							
G	119(77.78)	34(22.22)	1		1		
C	131(82.39)	28(17.61)	1.33(0.76–2.33)	0.308	0.34(0.09–1.21)	0.097	0.005
rs3184504							
Genotype							
CC	128(80.50)	31(19.50)	-	-	-	-	
Alleles							
C	256(80.50)	62(19.50)	-	-	-	-	
rs4149601							
Genotypes							
AA	34(85.00)	06(15.00)	1		1		
GG	27(77.14)	08(22.86)	0.88(0.34–2.29)	0.806	0.59(0.17–2.07)	0.417	0.024
GA	67(79.76)	17(20.24)	1.43(0.51–3.98)	0.485	0.41(0.96–1.74)	0.228	0.013
Alleles							
A	135(82.32)	29(17.68)	1		1		
G	121(78.57)	33(21.43)	1.27(0.72–2.21)	0.400	1.98(0.52–7.52)	0.312	0.018
rs4551053							
Genotypes							
GG	95(78.51)	26(21.49)	1		1		
AA	01(100)	-	-		-		
AG	33(86.84)	05(13.16)	0.55(0.19–1.57)	0.272	2.30(0.65–8.12)	0.714	0.042
Alleles							
G	223(79.64)	57(20.36)	1		1		
A	35(87.50)	05(12.50)	1.78(0.67–4.77)	0.245	0.17(0.01–2.00)	0.162	0.009
rs4791040							
Genotypes							
TT	44(86.27)	07(13.73)	1		1		
CC	19(73.08)	07(26.92)	0.87(0.32–2.15)	0.713	0.97(0.21–4.37)	0.970	0.057
TC	26(78.79)	07(21.21	1.42(0.53–3.75)	0.478	0.50(0.08–2.84)	0.437	0.025
Alleles							
T	114(84.44)	21(15.56)	1		1		
C	64(75.29)	21(24.71)	1.78(0.90–3.50)	0.095	0.10(0.01–0.60)	0.018	0.0007
rs5051							
Genotypes							
CC	-	-	-		-		
TT	88(82.24)	19(17.76)	1		1		
CT	09(90.00)	01(10.00)	0.88(0.34–2.24)	0.796	1.24(0.39–3.95)	0.714	0.042
Alleles							
C	9(90.00)	01(10.00)	1		1		
T	185(82.59)	39(17.41)	0.52(0.06–4.28)	0.549	1.90(0.15–24.22)	0.618	0.036
rs6083538							
Genotypes							
CC	94(82.46)	20(17.54)	1		1		
TT	02(50.00)	02(50.00)	1.44(0.59–3.51)	0.419	0.63(0.07–5.64)	0.686	0.040
CT	29(76.32)	09(23.68)	1.08(0.91–6.19)	0.926	0.99(0.13–7.56)	0.994	0.058
Alleles							
C	217(81.58)	49(18.42)	1		1		
T	33(71.74)	13(28.26)	0.57(0.28–1.16)	0.126	2.55(0.56–11.52)	0.221	0.013

HPT = Hypertension; dbSNP = single nucleotide polymorphism; CI = Confidence interval.

**Table 5 jpm-10-00267-t005:** Adjusted and unadjusted logistic regression models showing genotypes and alleles associated with blood pressure response to hydrochlorothiazide among Zulu and Swati patients (*n* = 131).

dbSNP	Uncontrolled HPT (*n*; %)	Controlled HPT *(n*; %)	Unadjusted Odds Ratios (95% CI)	*p*-Value	Adjusted Odds Ratios (95% CI)	*p*-Value	Bonferroni-Adjusted *p*-Value
All	71(54.19)	60(45.80)					
rs2070744							
Genotypes							
TT	53(55.79)	42(44.21)	1		1		
CC	02(40.00)	03(60.00)	4.22(1.15–15.47)	0.030	10.44(2.16–50.29)	0.003	0.0015
TC	16(51.61)	15(48.39)	0.10(0.02–0.48)	0.004	38.76(5.54270.76)	0.00023	0.0001
Alleles							
T	122(55.71)	99(44.29)	1		1		
C	20(45.55)	21(57.45)	0.77(0.39–1.50)	0.449	1.68(0.82–3.42)	0.151	0.076
rs7297610							
Genotypes							
CC	27(54.00)	23(46.00)	1		1		
TT	22(47.83)	24(52.17)	0.44(0.18–1.07)	0.07	2.28(8.55–6.11)	0.99	0.495
CT	21(61.76)	13(38.24)	0.40(0.16–0.98)	0.045	0.94(0.37–2.34)	0.898	0.449
Alleles							
C	75(55.97)	59(44.03)	1		1		
T	65(51.59)	61(48.41)	0.60(0.36–0.98)	0.043	1.86(1.09–3.14)	0.023	0.011

HPT = Hypertension; CI = Confidence interval, dbSNP = Single nucleotide polymorphism.

## Data Availability

The data presented in this study is available from the corresponding author upon reasonable response.

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
