# Peer review of "Genomic Association of Single Nucleotide Polymorphisms with Blood Pressure Response to Hydrochlorothiazide among South African Adults with Hypertension"

_jpm, 2020, doi:10.3390/jpm10040267_

Round 1
Reviewer 1 Report
The study titled: Genomic Association of Single Nucleotide Polymorphisms with Blood Pressure Response to Hydrochlorothiazide among South African Adults with Hypertension by Charity et al. evaluated the single nucleotide polymorphisms (SNPs) in hydrochlorothiazide associated genes and further assesses their correlation with blood pressure control among South African adults living with hypertension.
Specific comments:
- The authors mentioned YEATS4 some places YEAST4 some places, needs to correct proper terminology.
- The authors could have displayed the data in bar charts, at least important data related to allele to understand clearly.
- The authors should discuss the outcome of this study, the translatability of this work.
Author Response
Dear Editor,
Indeed, we are grateful to the insightful comments of the highly esteemed reviewers. Please find the responses to the reviewers’ comments. We therefore look forward to the final editorial decision.
Regards
Charity Masilela (for authors)
Reviewer 1:
African Adults with Hypertension by Charity et al. evaluated the single nucleotide polymorphisms (SNPs) in hydrochlorothiazide associated genes and further assesses their correlation with blood pressure control among South African adults living with hypertension.
Comment 1: The authors mentioned YEATS4 some places YEAST4 some places, needs to correct proper terminology.
Response: The term has been corrected.
Comment 2: The authors could have displayed the data in bar charts, at least important data related to allele to understand clearly.
Response: Thanks for this suggestion.
Comment 3: The authors should discuss the outcome of this study, the translatability of this work.
Response: Translatability of the results is discussed in the conclusion.
Reviewer 2 Report
COMMENTS
The article is devoted to the pharmacogenetics of arterial hypertension, in particular, to search for association of the effects of the hydrochlorothiazide (HCTZ) on the blood pressure with the polymorphic markers of number of genes, including PRKCA, WNK1, ADRB2 and NOS3.
The authors claim that there are at least four significant associations of polymorphic SNPs with the effects of hydrochlorothiazide (HCTZ) on the inclusion of hypertensive patients into one of two groups: with controlled or uncontrolled arterial hypertension.
The authors emphasize the influence of the ethnic origin of patients on both the SNPs polymorphisms and the presence of SNPs associations with the effects of HCTZ, which per se is of great interest for medical pharmacogenetics.
However, the assessment of the true value of these results is complicated by the fact that the studied patients, along with HCTZ, received treatment with other drugs (the authors do not list these drugs), which could affect the blood pressure and, consequently, the levels of SNP associations with the HCTZ treatment. How this contradiction may be resolved? In addition, the following questions arise regarding the mathematical and statistical analysis of the material.
- The authors use the adjusted odds ratios but do not explain what kind of covariates are introduced (line 149).
- The authors define the threshold of p-value as 0.05 (line 150). However, they test several independent SNPs, so Bonferroni correction must be used. For Xhosa sample the threshold p-value is calculated as 0.05/17 = 0.0029 and for Zulu and Swati sample this value is 0.05/2 = 0.025.
- It is unclear what means “expression of SNP” (section 3.2). The term “expression” refers to a gene, not to a nucleotide.
- Table 3. MAF defined as a minor allele frequency cannot be more then 0.5. The comparison of the frequencies of one of two alleles is more correct for populations where the minor alleles are different.
- Table 4. All associations are insignificant because all p-values > 0.0029 (see comment 2).
- Table 5. Two SNPs are significantly associated, but results for rs2070744 seem strange: the genotypes demonstrate the highly significant association while the alleles do not. I recommend to check calculations.
- The expression “using a whole genome approach” is incorrect (line 339). The authors used the gene candidate approach.
- P-values must be added to all OR in the text because the 95% confidence interval does not illustrate the significance of association when the multiple testing is performed.
Minor remarks:
- Define AOR and COR, please.
- Check all calculations. For rs4791040, OR = 0.99 but this value is outside the confidence interval (0.01 – 0.60). For rs7297610, p-value = 0.023 but not 0.019. For rs2070744, the numbers of alleles are incorrect in Table 5 and p-value = 0.00023 but not <0.0001.
- Letters in the names of genes (Table 2) must be italic
Conclusion: the authors have obtained extensive and potentially interesting material
Author Response
Dear Editor,
Indeed, we are grateful to the insightful comments of the highly esteemed reviewers. Please find the responses to the reviewers’ comments. We therefore look forward to the final editorial decision.
Regards
Charity Masilela (for authors)
Reviewer 2:
The article is devoted to the pharmacogenetics of arterial hypertension, in particular, to search for association of the effects of the hydrochlorothiazide (HCTZ) on the blood pressure with the polymorphic markers of number of genes, including PRKCA, WNK1, ADRB2 and NOS3. The authors claim that there are at least four significant associations of polymorphic SNPs with the effects of hydrochlorothiazide (HCTZ) on the inclusion of hypertensive patients into one of two groups: with controlled or uncontrolled arterial hypertension. The authors emphasize the influence of the ethnic origin of patients on both the SNPs polymorphisms and the presence of SNPs associations with the effects of HCTZ, which per se is of great interest for medical pharmacogenetics.
Comment 1: However, the assessment of the true value of these results is complicated by the fact that the studied patients, along with HCTZ, received treatment with other drugs (the authors do not list these drugs), which could affect the blood pressure and, consequently, the levels of SNP associations with the HCTZ treatment. How this contradiction may be resolved? In addition, the following questions arise regarding the mathematical and statistical analysis of the material.
Response: Many thanks for these critical questions. We have provided the list of the other anti-hypertensive drugs used by the participants. We indeed acknowledge the challenge of teasing out the effect of HCTZ in relation to the overall effect of all the anti-hypertensive drugs used by the patients. Notwithstanding, we focused solely on the associations between the SNPs and BP response. It should however, be noted that patients on sole treatment with HCTZ would be desirable for this study but this is rarely feasible; given that large proportion of patients are treated with combination therapy.
Comment 2: The authors use the adjusted odds ratios but do not explain what kind of covariates are introduced (line 149).
Response: Covariates have been explained in the methods section under statistical analyses
Comment 3: The authors define the threshold of p-value as 0.05 (line 150). However, they test several independent SNPs, so Bonferroni correction must be used. For Xhosa sample the threshold p-value is calculated as 0.05/17 = 0.0029 and for Zulu and Swati sample this value is 0.05/2 = 0.025.
Response: Thanks for this suggestion. We have corrected the text according to the reviewer’s suggestion.
Comment 4: It is unclear what means “expression of SNP” (section 3.2). The term “expression” refers to a gene, not to a nucleotide.
Response: We have corrected this to state “detection of a SNP”.
Comment 5: Table 3. MAF defined as a minor allele frequency cannot be more then 0.5. The comparison of the frequencies of one of two alleles is more correct for populations where the minor alleles are different.
Response: The table has been corrected to reflect only the MAF of observed SNPs in each population.
Comment 6: Table 4. All associations are insignificant because all p-values > 0.0029 (see comment 2).
Response: We have corrected the table according to the reviewer’s suggestion.
Comment 7: Table 5. Two SNPs are significantly associated, but results for rs2070744 seem strange: the genotypes demonstrate the highly significant association while the alleles do not. I recommend to check calculations.
Response: We have checked the calculations and they are indeed correct; however, we have added the following to the text to better define this phenomenon: “Although the AOR results show a significant association between the CC genotype and blood pressure response to hydrochlorothiazide, the large difference between the number of alleles observed discounts this significance. Given the large difference in numbers between the T-allele and C-allele, coupled with the uneven spread of genotypes at this locus, it may be suggested that these findings are not significant but are instead a product of skewed observations. Future study should aim for larger sample size to further examine this association.
Comment 8: The expression “using a whole genome approach” is incorrect (line 339). The authors used the gene candidate approach.
Response: The phrasing has been corrected to state: “candidate gene approach”.
Comment 9: P-values must be added to all OR in the text because the 95% confidence interval does not illustrate the significance of association when the multiple testing is performed.
Response: P-values have been added to all OR’s in the text.
Comment 10: Define AOR and COR, please.
Response: The abbreviations have been defined.
Comment 11: Check all calculations. For rs4791040, OR = 0.99 but this value is outside the confidence interval (0.01 – 0.60). For rs7297610, p-value = 0.023 but not 0.019. For rs2070744, the numbers of alleles are incorrect in Table 5 and p-value = 0.00023 but not <0.0001.
Response: Thanks for the suggested corrections. We have made the changes and all calculations have been verified.
Comment 12: Letters in the names of genes (Table 2) must be italic
Response: All genes have been italized both in table 2 and in the text.
Comment 13: Conclusion: the authors have obtained extensive and potentially interesting material.
Response: Thank you for your insight on our paper and the revisions have improved the quality of our paper.